# Policy Advocacy and NGOs Assisting Immigrants: Legitimacy, Accountability and the Perceived Attitude of the Majority

Agnieszka Zogata-Kusz 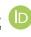

Department of Christian Social Work, Sts Cyril and Methodius Faculty of Theology, Palacký University Olomouc, 779 00 Olomouc, Czech Republic; agnieszka.zogata@upol.cz

**Abstract:** The article addresses the involvement of non-governmental organizations (NGOs) assisting immigrants in policy advocacy (PA) connecting the perspectives of political science and social work. In a context in which many politicians and a major part of society opposes immigration, it examines how NGOs perceive their legitimacy and accountability concerning their attempts to influence policymaking. It also studies how the attitude of the society towards these NGOs affects their work. The analysis builds on the multimethod research combining qualitative and quantitative approaches carried out among Czech NGOs. Among the key findings is that, when talking about legitimacy, NGOs' representatives refer mainly to themselves and their own vision of society. This is however a manifestation of *internalized external legitimacy sources* such as democratic principles and existing laws, together with experience and direct contact with clients, as well as moral obligations. As for accountability, despite many people identify these NGOs as irresponsible *welcomers*, in fact most of them feel accountable primarily to society (in particular its weakest parts), then to immigrants and finally to themselves. The negative attitude of the majority toward these NGOs clearly affects their PA activities, e.g., their access to authorities, the raised topics and applied tools or types of arguments.

**Keywords:** legitimacy; accountability; political participation; macro practice social work; civil society; Czech Republic; policymaking; integration policy; immigration policy; migration governance

## 1. Introduction

In many countries, non-governmental organizations (NGOs) function as service providers for various groups of society. Nevertheless, some of them additionally carry out policy advocacy (PA) trying to affect the making and enforcement of policies. Some parts of society in relation to that ask why the NGOs interfere with policymaking, what or who gives them the right to engage or who they feel accountable to for their actions. This regards, in particular, the situations when NGOs raise their voices in issues with which many people do not agree or are suspicious about. This article addresses the questions of legitimacy and accountability related to the policy advocacy activities carried out by NGOs—the service providers—i.e., questions that have not earned much attention in research. Focusing on the organizations assisting immigrants in one of the Central and Eastern European countries, it fills the gap in the current state of art (see below).

Non-governmental organizations (NGOs) working with immigrants in Czechia have played an important role in their integration for many years. Being service providers, many of them have also been engaged in policy advocacy concerning immigration and integration policies. When the subject of immigration became widely mediatized and politicized in relation to the crisis in migration policies in Europe since 2015, non-governmental organizations assisting immigrants consequently found themselves in a situation where some parts of society began to accuse them of attracting strangers to the relatively homogeneous society.

In such a context, this article challenges three questions. The first, how these NGOs define sources of legitimacy for their policy advocacy (PA) involvement. In other words,

who or what gives them the right (according to their perception) to raise a voice? The second, who they feel accountable to in relation to these activities. The third, how they perceive the attitude of Czech society and politicians towards themselves and how it affects their PA work.

The examination of these questions is based on multimethod qualitative research combing the content analysis of NGOs' websites, a focus group and individual semi-structured interviews. Additionally, quantitative research questionnaire enabled answering the question regarding the perception of the attitude of the environment towards the NGOs assisting immigrants. The concepts of policy advocacy, legitimacy and accountability provide a theoretical basis for the analysis. The article connects the perspectives of two disciplines: political science and social work.

The importance of the above-mentioned questions relates to at least two issues. The first is the transformation of the civil sector and concerns about whether NGOs are guardians of democracy or advocates of their own ideas and interests who are detached from reality. In this context, questions regarding NGOs' involvement in influencing policy, their legitimacy and accountability are crucial. The second point that makes the topic essential is the systematically growing number of migrants—not only in Czechia, but in other European countries, including those for which this is still a relatively new phenomenon (e.g., Central and Eastern European countries). Consequently, in particular there, the NGOs' attempts to influence policies concerning immigrants may raise concerns in some parts of those societies. We will develop these issues in the following paragraphs.

### 1.1. NGOs' Policy Advocacy in Research Studies

Whereas the topic of NGOs' policy advocacy activities has been examined several times already (Nicholson-Crotty 2007; Albareda 2018; MacIndoe and Beaton 2019), the engagement of service providers in PA is relatively seldom a subject of research studies. It occurs mainly in social work studies (Mosley 2010; MacIndoe and Whalen 2013; De Corte et al. 2021). In recent years, specifically with the involvement of NGOs assisting immigrants in PA, the topic has occurred in several researches (Ambrosini 2013; Chin 2018; Follis 2019; Trei 2021; Calderon et al. 2021); however, none of them delved very deeply into the question of legitimacy and accountability associated with this activity. Ambrosini (2013) explored the motivations, strategies, coalitions, and outcomes of the activities of two specific Italian NGOs. Chin (2018) similarly focused on the questions of motivations, strategies and alliances of a specific NGO's coalition activity in New York City. Follis (2019) analyzed the relationship between the Law and Justice party ruling in Poland (refusing refugees) and civil society organizations advocating for refugees' rights. Trei (2021) attempted to explain the involvement of Estonian NGOs in asylum policy decision making on the national level. Calderon et al. (2021) addressed the general questions of engagement of American immigrants assisting organizations in policy advocacy (mainly if they engage, when yes, and when no). Thus, the presented study complements the understanding of NGOs involvement in policy advocacy. It fills in the gap with the findings focusing on legitimacy and accountability related to PA carried out by service-providers in a hostile environment in one of the CEE countries.

### 1.2. Theoretical Context: Policy Advocacy, Legitimacy and Accountability

Policy advocacy, legitimacy and accountability are the main theoretical concepts of the article. Policy advocacy is often referred to as "any attempt to influence the decisions of any institutional elite on behalf of a collective interest" (Jenkins in Mosley 2010, p. 58). Almog-Bar and Schmid (2014) were more specific concerning the directions and addressees of policy advocacy endeavors. The authors stipulated that they were undertaken for a collective goal to bring about, affect or prevent policy changes, and decisions made by politicians, government or authorities at various levels, including the local one. They may concern both policymaking and policy enforcement, thus, the actual practice.

These definitions come from the social work discipline and its branch—macro practice social work. Given that, the service organizations are specific types of NGOs providing legal and social counselling and carrying out social work with their clients, referring to the social work definition is appropriate. Political science does not use the term 'policy advocacy'. While talking about NGOs and interest groups it refers to 'political participation' which is, however, a much broader concept (e.g., see definitions in Ekman and Amnå 2012).

Policy advocacy relates in this article to a wide range of activities (including lobbying), supporting rights and promoting the interests of immigrants and the foreign-born. It regards the area of legislation (and legal regulations at other levels than the national one), but also more broadly it concerns the process of formulating policies together with their implementation and enforcement at various levels.

Engaging in policy advocacy, i.e., interfering in policymaking, often raises questions about legitimacy and accountability of those who engage in it.

Despite the notion that civil society and democracy are essential for each other, the fact that organizations attempt to influence policymaking frequently brings about the question of their legitimacy to do so. NGOs are required or expected to justify their entitlement to engage in policy advocacy: who voted for them? Why are they trying to interfere in the policy process? What gives them the right to do so?

The concept of legitimacy originally regarded the political regimes and the state authority (see Weber 1947), nonetheless, for many years already it has been used in relation to NGOs activities. Various authors have defined legitimacy as the right to do something, as the recognition that one's actions according to the rules, norms, beliefs, and values of the society (Brown and Jagadananda 2007; Edwards 2000). For the purpose of the article, we accept Suchman's definition who understands legitimacy as "a generalized perception or assumption that the actions of an entity are desirable, proper, or appropriate within some socially constructed system of norms, values, beliefs and definitions" (Suchman 1995, p. 547). This definition corresponds to the common understanding that legitimacy is always a mandate that the stakeholders confer to the organization. In particular, the neoinstitutionalism school points at the organization's environment as the legitimacy source (Brinkerhoff 2005, p. 5). Nonetheless, even though we accept this assumption, we recognize the NGOs seeking legitimacy are not passive elements of the system. Puljek-Shank noticed that legitimacy arises in 'legitimation' processes in which civil society organizations actively make claims of legitimacy and, in such a way that they may affect their environment (Puljek-Shank 2018). As Gnes and Vermeulen noted, the concept of legitimacy and legitimation are valuable since "they bring to light the processes through which organizational entities justify their right to exist and their actions within a particular normative context" (Gnes and Vermeulen 2019, p. 218). It is the NGO's task to keep explaining to its environment what its mission and activities are. For each organization, a basis for sending signals to the environment regarding the legitimacy is making it clear to itself, why it has the mandate to do what it does (or wants to do). This idea lays behind the research questions of this article.

Next to legitimacy, another issue that raises doubts is accountability: to whom the NGOs feel accountable for their PA activities and their potential results. Are they donors? Immigrants? Taxpayers? Such questions emerge particularly when somebody calls for policy changes that the great majority of public opinion opposes. Immigrants rights advocacy seems to be a striking example in today's Central Eastern Europe. Even though the public often do not know what the exact subject of the PA activities is.

There are many definitions of the concept and authors distinguish numerous types or forms of accountability taking into consideration various criteria. For instance, Brown and Jagadananda defined accountability as "a responsibility to answer for particular performance expectations to specific stakeholders" (Brown and Jagadananda 2007, p. 9). Edwards and Hulme understood it specifically in their publication from 1994 as "the means by which individuals and organizations report to a recognized authority (or authorities) and are held responsible for their actions" (in Edwards and Hulme 2014, p. 9). In this article, we follow a stance of Unerman and O'Dwyer (2006) who theorized accountability in the

context of NGO advocacy. The authors differentiated two types of accountability. The first one is *relational accountability*, for which the identification of stakeholders is a vital point since policy changes that advocacy activities may bring affect various groups to a different extent (Unerman and O'Dwyer 2006). Some believe that actors should recognize their duty of accountability towards all people whose life the advocacy's actions may influence (Unerman and Bennett 2004). The second type refers to a completely different understanding of accountability: it is about feeling the duty to follow an NGO's values, missions, and beliefs, and not to the subjects that any policy changes could affect. Unerman and O'Dwyer call this *identity accountability* (Unerman and O'Dwyer 2006, p. 356). Taking the Unerman and O'Dwyer's concepts as a starting point, this study examined which of them were accepted bythe immigrants assisting organizations in Czechia.

### 1.3. Czech Context

Civil society organizations are "commonplace in advanced democracies" (Frič 2016); however, there are concerns that NGOs, influenced by the modernization processes of professionalization and the individualization of civic participation, are effectively moving away from society (Frič 2016) and therefore (perhaps) are not able to fulfil the role of *transmission belts* for connecting citizens with policymakers, transferring the preferences of one to the other (Albareda 2018). Frič even talks about "the paradoxical phenomenon of NGOs uprooting from public life of citizens" (Frič 2015, p. 10). This phenomenon particularly negatively affects their advocacy capacity and challenges their legitimacy as platforms for civic participation and it is not typical only for Czechia, e.g., as in neighboring Poland, where Korolczuk notes it entered the post-NGO age "epoka postengieosowa" in which associations and foundations are no longer the dominant form of self-organization (Korolczuk 2017, p. 2).

In Czechia, the discourse on the role of NGOs in society dates back to the 1990s, when civil society organizations began to emerge after the fall of communism and the turning to the West. Ex-President Václav Havel recognized and promoted the idea of citizens' engagement in a range of activities, including political ones related to public affairs (Havel 1999). Havel's notion of democracy follows the Dahl's concept that recognizes the important role of civic engagement for fulfilling democratic criteria, e.g., effective participation, enlightened understanding, control of the agenda and inclusiveness (Dahl 1989). Havel saw civil society and civil society organizations as the backbone of democracy. By contrast, President Klaus was much closer to Schumpeter's minimalist conception of democracy, which emphasizes the importance of elections and political parties. Klaus did not acknowledge the possibility of any influence on politics by non-elected actors. On the contrary, he warned of "NGOism" and accepted John Fonte's metaphor when referring to nongovernmental organizations as democracy's Trojan Horse. He pointed out that they were not elected by anyone and were only accountable to themselves (Klaus 2005).

The discourse between the two presidents has been influencing the attitudes of some politicians towards NGOs and their statements about them and has spilled over into society. Especially in the context of the migration policy crisis around 2015, some politicians started to attack NGOs (in general, not only those assisting immigrants). In June 2016, President Miloš Zeman referred to NGOs as leeches on the state budget and Prime Minister Andrej Babiš repeatedly spoke about the need to reconsider subsidies for non-profit organizations (Guryczová and Kočí 2018).

The 2015 crisis came at the moment when the Czech government led by Bohuslav Sobotka consisted of the centre-left Czech Social Democratic Party—ČSFD, the populist party ANO 2011 (in English: YES), and the Christian and Democratic Union—the Czechoslovak People's Party. Andrej Babiš, the leader of ANO 2011, was the First Deputy Prime Minister and the Minister of Finance. As a result of a range of scandals, Babiš was forced to resign his functions in May 2017. Despite that, in October 2017 he won along with his party the Parliamentary elections. His one-party government, however, failed a confidence vote in the Chamber of Deputies of the Czech Republic. Eventually, thanks to the support

from President Zeman, the cabinet ruled as the government in resignation until June 2018, when Babiš formed a new government consisting of the ANO and ČSSD with the support of the Communist Party of Bohemia and Moravia. Zeman has been the Czech president since 2013. In 2018, he was re-elected for the function.

It was at that time when NGOs and the people associated with them started to be pejoratively called 'welcomers'. Society's trust in NGOs had significantly decreased (although it was not very high before). Since the middle of 2015 it maintained below 40% trust (e.g., in October 2017 only 32 % of Czechs trusted NGOs), while distrust clearly increased, being at a level above 50% (in October 2017, 56% of Czechs did not trust NGOs, and in September 2019 it was up to 58% (CVVM 2019). We must be aware, however, that this number is not telling, because it speaks about NGOs in general, without distinguishing their type, function, or focus, and thus we do not know what the respondents imagined of 'an NGO' answering to the question regarding trust.

The second thing that underlines the importance of the issues addressed here is the growing number of foreigners in CEE countries and the related need to integrate them into receiving societies, to have them focus on immigration and integration policies and related policy advocacy. Currently, the Czech Republic is the country with the highest number of immigrants within the Visegrad Group. Immigration to Czechia has been growing for many years with approximately 30 thousand–40 thousand per year. At the end of 2020 it reached almost 633 thousand which represents 5.9 percent of the entire population—which counts 10.7 million people (CZSO 2021).

The state has been developing its policy towards immigrants since the end of the 1990s to support their integration with the receiving majority and to prevent negative social phenomena (see, e.g., Zogata-Kusz 2017, 2020). Non-governmental organizations became its supporters in these endeavors providing services to immigrants and their families. They offer foreigners social and legal counselling, Czech language courses and courses on social and cultural orientation. Similarly, they work with the receiving majority creating opportunities for meeting and cooperation between Czechs and immigrants and raising the majority's awareness about integration-related issues. Currently many NGOs are highly professional organizations. They run projects oriented also on more specific questions such as increasing immigrants' parental competences (the Centre for Integration of Foreigners), socially excluded migrant women (Association for Integration and Migration) or teaching journalists how to cover migration-related topics (People in Need). Most NGOs cooperate with municipalities, regions and the national authorities trying to not be merely simple assistants to the state, not only service-providers, but valuable partners for policy-making—in some cases also at the level of the European Union (see further). It was only in 2009 that the Czech Ministry of Interior established state integration centers that offer similar services and since 2021 have been organising short obligatory adaptation and integration courses that newcomers have to attend within one year of the date of collecting their residence permit.

Migration has become one of the main media and political topics since 2015. Some parties have tried to win the support of the electorate in their election campaigns by raising anti-immigration slogans. Anti-immigration rhetoric was an important factor of the election success of the far-right party, Freedom and Direct Democracy, led by Tomio Okamura. Even Miloš Zeman won the second round of the presidential election with the slogan, "*Stop to immigrants and Drahoš* [i.e., the other candidate]. *This is our country. Vote for Zeman!*", suggesting that the other candidate welcomes refugees.

The narrative of Czech politicians and the media coverage of migration has influenced Czech society. For example, in 2017, as many as 81 percent of Czechs had a negative attitude towards it. Indubitably, the opinion poll data has serious limitations, do not reflect the full picture, and lack nuance. Despite that, in a comparative perspective of other societies, we may see that the approach of Czech society to the immigration of third-country nationals is one of the worst among the EU member states (Eurobarometr 2017, p. 112).

The negative narrative associated with immigration is, however, present in many European countries. Numerous politicians—from not only far-right political parties—the media, and various other groups use immigrants or refugees as scapegoats and refugee immigration (potential) as a pretext to introduce policy restrictions or as a smokescreen to cover important social, economic or political problems (see, e.g., Bocskor 2018; Arcimaviciene and Baglama 2018; Harraway and Wong 2021). They spread not only the division of 'us' and 'them' but frequently develop hate speech which may lead to discrimination, racism and hate crimes. Boréus talks in this context about the "threat perspective on refugee immigration" and the "native dominance perspective on integration" (Boréus 2021). In consequence, this kind of atmosphere hampers the work of NGOs assisting immigrants, and in particular their policy advocacy.

## 2. Four Stages of Multi-Method Research

This presented study used an empirical-analytical approach and specifically a case study method. As for data collection, the article reports the results and conclusions from this multi-method research that took place from 2017 to 2019 in Czechia. It is the first research aimed at NGOs assisting immigrants which are active in policy advocacy in this country. The research consisted of four stages. The first, the preparatory stage, was the identification of the service-providing organizations active in policy advocacy at various levels. This was possible thanks to an analysis of their websites content. It followed, if an NGO were active in PA, what it does in this area and at what level (UN, EU, national, regional, local). I also looked for signs of claims on their legitimacy and accountability in PA which provided me some of the introductory material for the formulation of questions for later research phases. Altogether, I went through the websites of 20 organizations—members of the so-called migration consortium (The Consortium of Migrants Assisting Organizations), which is associating with the main organizations working with immigrants in Czechia. The core criterion to consider with the NGO in this research was that it was a service provider and that its main target group (client group) were immigrants, i.e., people with a migration background, regardless of whether it was so-called voluntary or forced migration (for some consortium members the predominant activity was public education, e.g., Migration Encyclopedia or the Multicultural Center Prague). Within the first stage, I also carried out preliminary consultations with NGOs' representatives and I participated in the V4 Regional Migration Alliance Conference organized in Bratislava in 2017 by Oxfam International, which dealt with NGOs' cooperation with governments. The second stage was to carry out the focus group interview. Eventually, I decided to invite the advocacy group of the Consortium of Migrants Assisting Organizations to participate in the research. The migration consortium is an umbrella organization for Czech organizations engaged in migration issues. The consortium itself is small. An important matter is, however, that representatives of its member organizations (mainly service providing organizations), gathered in a so-called advocacy group to carry out a vast part of its advocacy work (see below). Altogether, the representatives of nine organizations took part in the focus group in Prague. They were either NGOs directors, or regular employees with experience in PA. The main questions asked during the focus group were: 1. *From where do you draw legitimacy for your advocacy activities, in other words, how do you perceive who or what gives you the right to try to influence the policy?* 2. *Do you feel accountable to anyone for your advocacy work? and to whom?*

The third stage was a set of six individual semi-structured interviews with people involved in PA. They focused on the clarification of the types of PA activities that the NGOs carry out, the argumentation that they use as well as the relationships among the NGOs assisting immigrants and the possible impact of their PA. Finally, their content regarded also the perceived relationship of the society towards these organizations since some of the comments during the focus group indicated that there could be a problem affecting NGOs' policy advocacy and the way of thinking about it. Most of the interviewees represented organizations that had participated in the focus group interview (including the policy

officer of the consortium). The exception was the NGO, Most PRO, whose representative, however, regularly participates in the advocacy group work as well. I arranged that in both cases, i.e., the focus group interview and the individual interviews, there were representatives of NGOs operating at various, i.e., local, national, and international levels, and sometimes at all of those levels (e.g., SIMI—Association for Migration and Integration or OPU – Organization for Aid to Refugees). To ensure the ethical standards of the research, I guaranteed to the interviewees participating in the focus group as well as in individual interviews that their answers would stay anonymous.

After the data collection, the content analysis of the gathered material followed. After each stage separately, I carried out coding of the material, assigning categories and interpreting the data. In this article, quotations from individual interviews are labelled as IA, IB, IC and so forth, whereas the quotations of focus group participants are labeled as I1, I2, I3 and so forth.

During the interviews, I found that some people working in the NGOs assisting immigrants were often afraid to admit where they work while talking to other people. For this reason, I decided to broaden the research. To complement the data in relation to the third research question, I expanded the study and engaged an exploratory sequential design (Dawadi et al. 2021). The data gathered from the focus group and interviews allowed me to build a quantitative instrument to test the findings regarding the perceived attitude of Czech society and politicians towards the NGOs and its effect on their PA work. Concerning the integration of the research design, first it was present when the qualitative results became a foundation for the development of the questionnaire, and second, when I made integrated conclusions based on qualitative and quantitative data

The questionnaire combined open and closed questions distributed to people working with immigrants in NGOs. It became the fourth stage of the research. Its aim was to find out what reactions people working with migrants in NGOs encounter in the context of their work and what impact they have on their work. The questionnaire examined the perceived attitudes or reactions of people belonging to three circles of the NGO employees' social environment, e.g., the Czech society in general (thus the most external circle), strangers (thus the middle circle) and the close people (thus the circle that is the closest to the person). I addressed individual workers of nine service-providing NGOs, writing to their email addresses published on the organizations' websites asking them to fill in the questionnaire. The organizations have their branches in the whole of Czechia. The population comprised of appx. 170 people (the number was calculated based on the contact details available on the organizations' websites; the exact number is difficult to determine due to the high turnover of staff in some NGOs and, they do not always update addresses on the websites). It consisted of people with various expertise engaged mainly in legal and social counselling for immigrants (e.g., social workers, lawyers, psychologists). I excluded intercultural workers from the sample since most of them have a migration background themselves and this fact could have biased the results. Eventually, out of 63 collected questionnaires, I ignored four of them since people not working directly with immigrants completed them. I therefore analyzed a set of 59 questionnaires. One may estimate that this meant data from about 35% of the statistical population as a whole (the return rate of questionnaires from different organizations varied). The research was of an exploratory character, therefore, the data was sufficient despite not being reliable from a statistical point of view.

## 3. Results

The Czech Republic was the first country in the Visegrad region to introduce an integration policy (Zogata-Kusz 2020). The government accepted 'The Rules of Foreigners' Integration Concept' in Czechia as far back as 1999 and the first concept a year later (it is regularly updated). Since then, there has been an action plan each year for the concept realization. Those who implement the plan are largely NGOs receiving public grants for their tasks (although it is not their only financial source). This helped the Czech NGOs assisting migrants to develop (even though some of them had existed since the beginning

of 1990 when they were helping refugees from the Balkans). Already in 2003, they had established the aforementioned umbrella organization called the Consortium of Migrants Assisting Organizations. The consortium's goal is to facilitate cooperation among the organizations, to increase the dialog of the actors with various approaches to migration issues, to engage in policy advocacy to provide feedback to policymakers, and to ensure just integration and immigration policies. There are a few working groups in the consortium, e.g., legal, social, media and policy advocacy. The consortium is involved in PA at various levels, including the international level. An example is their activity in the Platform for International Cooperation on Undocumented Migrants (PICUM), through which they actively participated in work on the Global Compact for Migration. Simultaneously, the consortium member organizations separately develop advocacy activities on a number of platforms, often next to the consortium (e.g., Association for Integration and Migration). Reflecting such situations, ID noted that "*the relationship is complementary*." The consortium fulfils a networking and coordinating role. It is similarly attempting to harmonize various perspectives among its members by "*trying . . . to find some consensus on what we're going to say, and not to have any major contradictions between how the member organizations are approaching . . . key policy issues, because that would weaken the whole sector of course*" (IA).

The consortium and several of its member organizations have relatively extensive experience in PA and they use a range of advocacy tools and methods. For instance, they produce and spread analyses, policy papers, factsheets and booklets (e.g., regarding problems with foreigners' health insurance in Czechia or the possible impacts of proposed changes in the *Act on Aliens*). They also organize meetings with academics and policymakers or participate in various bodies dealing with immigrant issues (e.g., the governmental Committee on the Rights of Foreigners, and the municipality level commissions for community planning of social services). To increase the chance for the required changes, they address members of parliament, other policymakers and officials at various levels with their comments and proposals, they comment on amendments to laws and use litigation. Sometimes the individual NGO uses these tools regardless of the activities of the migration consortium. Sometimes, however, they make, e.g., analysis (or parts of it) for the consortium (as their members). Cooperation among the member organizations works well, e.g., regarding the comment procedure: people from various NGOs work out comments to bills. When it is needed and possible, the NGOs assisting migrants—or the consortium as whole—build coalitions with other NGOs (e.g., focused on rights of women) or other social partners (e.g., trade unions). Many—especially smaller—NGOs do not have the capacity to develop their own advocacy activities. In such cases, however, they support the consortium providing it, e.g., the evidence or case studies illustrating the problem. We should stress, however, that even these smaller organizations often do engage in advocacy work at a local or regional level, but they are reluctant to call it 'policy advocacy'. They treat it as "*normal cooperation*" (IC) with local policymakers. IC explained: "*we try to . . . help the municipality with our knowledge and experience from practice . . . this is a necessity for our work to work . . . we try to supply them with some information, [they] just know about our work . . . so that they can consult with us when they have problems.*"

Even though the organizations cooperate on PA issues, there are many differences among them. As ID said when asked about relationships among NGOs, "*it is a mix of competition and collaboration, depending on both with whom and on what topic.*" There is competition for financial resources; there are ideological differences or a diverse perception of the most appropriate working methods. Yet, the NGOs agreed that there is a great deal of cooperation and solidarity among them, mainly when it comes to PA. As IA noted, "*the very existence of the consortium is an expression of a kind of will to cooperate.*" If any disagreements appear, different opinions concerning how to address some problem or which problem to focus on, then the organizations try to solve it internally, balance various perspectives within the consortium and then speak with one voice externally. As I5 admitted: "*we are not fundamentally monolithic . . . and that we have . . . value clashes. That's what the debates are about and from that the external view and the process is catalyzed.*"

The Czech migration consortium is currently a leader in immigrants-related policy advocacy activities in the Visegrad region. In autumn 2017, the consortium developed the advocacy framework. The following sections provide answers to the main questions of the study.

*3.1. Legitimacy*

The analyses of the data collected in the multi-method research led me to point out that the core source of legitimacy for Czech NGOs, relevant for their policy advocacy activities, were principles related to civil society and democracy. As a part of civil society, the interviewees perceived themselves as *"co-responsible for the state"*, as the ones helping *"to cultivate [democratic] environment"*, *"to maintain liberal order."* They referred to civil and human rights, a freedom of expression in public matters as well as protection of the weaker as the values they identify with, and the values in which they see their legitimacy. As I7 said: *" . . . the loss of some people's rights will eventually fall on the heads of the majority that thinks that it is not affected, or, on the contrary, that it is protected by some policy"*.

Interviewees directly linked these views to the existing law as the source of legitimacy of NGO PA activities (*" . . . legitimacy is in the law"* as I5 put it). The law mostly contains principles that are in line with the interviewees' ideas such as the human-rights-related commitments of the state. These are, e.g., the 1993 Constitution of the Czech Republic, and the 1993 Charter of Fundamental Rights and Freedoms belonging to the Czech constitutional order (not to be confused with the EU Charter of Fundamental Rights). These commitments are similarly part of international legal obligations. In the law, the interviewees not only see the course of legitimacy for their PA engagement in general, but also the essential argument they use while advocating for immigrants' rights. They point out that the main problem is not usually the non-existence of certain norms, but rather their insufficient enforcement in practice. They refer to the idea of civil society as the watchdog of democracy and what they are afraid of—observing the developments in the Visegrad countries—is that the state will treat them only as service-providers needed for outsourcing services that it cannot deliver or does not want to deliver. While talking about civil society, democracy and the legal commitments, the representatives of NGOs present their ideas about the principles that are fundamental for a society to function. This is their notion concerning the society in which they would like to live.

As for other sources of legitimacy, it is challenging to determine their order of importance. The NGOs recognize that their knowledge gained through experience from work with clients gives them the right to engage in policy advocacy. IA noted that *"we have the know-how that nobody else has, even the state administration doesn't have it, because they just take a top-down approach to these people."* At the same time the desperation of people, which tends to repeat, makes the NGOs feel responsible for the change. As I1 said, e.g., *"we have a lot of people who just sit in our waiting room every day. We know their problems . . . who else but us should be trying to help refugees."* Referring to their own experience, the NGOs point at their attempts to seek evidence—or experience-based policymaking. To increase their credibility, when it is possible, they (e.g., People in Need) cooperate with universities, research institutes or employ researchers, usually in relation to specific projects, to gain hard data for their PA activities. This is, however, a rather extraordinary situation because of the lack of capacities. They mainly rely on their own expertise. Yet, as IB noticed *"it's a bit based on the principle that [the Ministry of] Interior applies . . . Unfortunately, [what they do] it's usually not backed up by any statistics or analyses, when even sometimes something big changes."*

As for the clients themselves (as a possible source of legitimacy), I asked if the NGOs *speak with* immigrants, *speak on behalf of* immigrants or *speak for them* (using the terms of Slim 2002). The interviewees mainly agreed that clients could not be a legitimacy source of an NGO's engagement in PA. *Speaking with* immigrants is problematic and challenging. Involving foreigners in PA endeavors depends on the concrete issue (to some extent it was possible, e.g., in the case of the campaign for the inclusion of foreigners in the general health insurance system). Some NGO representatives perceive a too small participation of

immigrants(clients) as a weakness in their activities. A possibility could be the cooperation with immigrant organizations; however, these are not politically active (IA: "*There are not that many active [immigrant] organizations who would involve*"). Interviewees definitely agree that it is impossible to say that they speak on behalf of immigrants. The most vital reason for this is the lack of mandate from their clients. Clients mainly do not participate at any stage of PA activities. Moreover, they are a very heterogeneous group, coming from various cultures and they bring from their home countries particular perspectives on socio-political issues. Even though we must not generalize, for some of the immigrants it is normal that a foreigner is supposed to have fewer rights but as I9 noted, "*they take it as legitimate and are only surprised when it hits them.*" Furthermore, solidarity is missing among the various immigrant groups and as I7 said, "*certainly not everyone respects some of the liberal ideas, they do not respect the equal rights of men and women, we definitely fight against [attitudes and opinions of] those migrants.*"

In the context of this part of discussion, some interviewees reflected on the question regarding who *a foreigner* actually is. From the legal point of view, it is a person without Czech citizenship. In reality, as the interviewees pointed out, policies often negatively affect many *citizens*—family members or people who have received Czech citizenship, but their skin color, name or birth certificate *make* them foreigners again. Interviewees therefore observe that the *themes* raised in policy advocacy are more important than *being immigrants' advocates*. Many topics, such as pensions or conditions of employment, concern the weak regardless of whether they are immigrants or not. For this reason, the interviewees perceived it as more appropriate to say that they *speak for* foreigners *and simultaneously for* the receiving majority.

Some interviewees brought different ideas about legitimacy, although definitely still related to their notion of the functioning of society in the first place. As another legitimacy source they see moral obligations, the ethos of the organization, religious obligations or, as I4 put it "*a moral obligation to our founding Chartist mothers*[1]".

Finally, regarding the topic of legitimacy, the interviewees perceived it as unjust and "*absurd*" that they are "*ostracized as an illegitimate force in advocacy*" (I6), since they try to be very transparent and clear about what they lobby for and why. They do so not only in relation to the policymakers but also to the public, e.g., providing interviews to the media, publishing their statements—including the *2015 Migration Manifest*, or carrying out public advocacy through campaigns and on public oriented projects. A few of them (e.g., IA) acknowledged that a great shortage of the current situation is the lack of any act on lobbying, thus, an act regulating issues of involvement in policymaking (the Czech Republic does not have any yet). This could explicitly give NGOs a mandate for lobbying under specific conditions.

*3.2. Accountability*

Interestingly, there was less accordance among the interviewees regarding accountability for their policy advocacy activities, specifically as for the order of importance for particular stakeholders. Even though the interviewees felt accountable, e.g., towards their clients, for only one of them was it the primary group. Most interviewees agreed that they were accountable towards the society and, in particular, towards specific groups of society that the required changes could affect most profoundly, or the groups missing potential changes. Some specified that they understand the majority society as being not only the groups they cooperate with (such as officers or teachers), but more broadly as the weakest parts of society, e.g., the unemployed or employees working in low job positions. A critical matter that a few interviewees agreed on was discussing and reflecting on the possible impacts of potential policy changes, not merely for immigrants but for the wider society and its specific groups. In this context, it is important to underline that the members of the advocacy group were very diverse as for their political views. They were aware that some policy changes may bring an additional burden, e.g., to the richer part of society but, as I7 admitted, "*the party that will defend the interests of the richer, stronger, etc., is . . . incomparably*

*stronger.*" I4 noted, "*It's not that we just say: OK, let's abolish the borders here, whatever happens, we don't care. The things that we're really fighting for are the things that we perceive are good for that society, and let the other* [i.e., the richer, the stronger] *side arrange themselves accordingly.*" While talking about accountability the interviewees again stressed the need for balancing various interests: accountability towards immigrants and human rights commitments as well as accountability towards various social groups or the majority. One interviewee (I5) compared it to the issue of the death penalty, which is not present in the Czech legal system despite most Czechs having supported it for a long time. Balancing interests and looking beyond the impact of their advocacy priorities are two extraordinarily important things for them.

Some NGOs' representatives pointed also at themselves, their relatives, and friends but also people from the consortium, as those to whom they are accountable for their activities, since PA may affect their image and credibility. Interestingly, some interviewees also mentioned concrete historical figures (such as Masaryk, Havel, signatories to Charter 77) and their legacy, and they referred to the Czech democratic tradition.

### 3.3. The Attitude of the Majority and Politicians towards NGOs Assisting Immigrants and Refugees

As mentioned above, the sources of organizational legitimacy lay outside the organization(s), and it is the environment that assumes its actions as right. For this reason, I wanted to find out how the representatives of the NGOs assisting immigrants perceive the attitude of the majority and politicians towards them. Firstly, despite the fact we know the general level of trust in the NGOs in Czechia, the opinion poll data is highly limited and mainly it does not distinguish between the trust towards various types of NGOs. Secondly, the perception of the majority's attitude towards the NGOs assisting immigrants (i.e., themselves) may affect their own legitimacy claims as well as their policy advocacy activities.

In the interviews, the NGO representatives admitted that they felt that Czechs do not understand their role and actual responsibility. This is to some extent a legacy of the former Czech president Václav Klaus's attitude towards civil society organizations and his understanding of democracy. As the interviewees observed, it currently relates to Miloš Zeman, the president, and Andrej Babiš, the prime minister (until December 2021), who have developed a negative narrative about NGOs and have related them closely to the acceptance of unwanted immigration. Many interviewees agree that the expression, *NGO*, has become "*a rude word*" (IB). The overall atmosphere around the NGOs in Czechia is not positive. As IA said, "*what we seem to be struggling with is pretty much what the whole sector is struggling with.*" In this uneven debate, the NGOs attempted to maintain their political relevance despite many politicians who would rather see them as simply service providers, a silent support for the state. IA admitted: "*I feel enormous pressure . . . maybe not immediately yet, but systemically, to just tie up the non-government . . . and actually sort of cut the political one . . . I think that's what irritates these people the most. That we're politically relevant . . . And at the same time, it's something that's just been the alpha and omega of our work for years. So we just have to maintain that political relevance.*"

The attitude towards the organizations assisting immigrants is especially adverse. Many politicians have misused immigration and asylum topics since 2015. This is not only a Czech problem but the problem of the whole region. IA referring to how Budapest and Warsaw limited the work of NGOs assisting immigrants said: "*what is happening in Hungary and Poland is simply an extreme warning for us: an extreme exclamation mark.*" The interviewees observed that a large part of society has become obstinate and deaf to any argument regarding foreigners and the situation has been worsening. As ID observed, "*we are perceived by these politicians, but also by the public, as someone who is actually profiting from the situation, who is untrustworthy.*"

Many politicians and a great majority of the public are not aware of what the NGOs' expertise is, or what their real attitudes are. Moreover, anti-immigrant political parties, such as, Freedom and Direct Democracy (Czech abbr. SPD), present themselves as the

only actors who are concerned for the good of Czechs. Several participants (e.g., ID, IE) admitted that the situation has escalated to the extent that they prefer not to tell strangers that they work in NGOs and that they work in particular with foreigners. This was an impulse for the further research. Although its results do not have fully corresponding value, since the quantitative research was of an explorative character and despite the attempt of representativeness, the sample was not representative (as mentioned above), they do provide a useful insight regarding the observed attitude of the environment towards the NGOs assisting immigrants.

The questionnaire distributed among employees of the NGOs helped me to discover that as many as 75% of respondents observed that after 2015 the public attitude towards NGOs assisting immigrants has worsened and it has become negative. Nonetheless, the picture is different when we look at the answers that the respondents gave on the question regarding their direct experience with the reactions of strangers (i.e., in the case of direct contact with people they do not know). The percentage of those who *mostly* come across *negative* or *rather negative* reactions related to the fact that they work in an NGO assisting immigrants is much smaller: 37%. This means it is about the same as for those who come across mainly *neutral* reactions (almost 38%) but more than those who mainly come across *positive* or *rather positive* reactions (almost 28%). When we look closer to other answers of those 37% of respondents who came across mainly negative reactions, we find interesting relations. As many as 91% of them face (*sometimes* or *often*) critical comments against immigrants, 77% of them meet critical comments against NGOs in general and 36% encounter critical comments towards themselves (for their work in an NGO assisting immigrants). At the same time, 18% of them answered that they do not face critical comments regarding themselves at all. These findings are interesting. Despite 75% of all respondents perceiving the majority's attitude towards NGO assisting immigrants as negative, only 37% of them perceive the reactions of strangers related to their work in such an NGO as negative, while only 36% of this specific group simultaneously pointed out that they had faced negative comments against themselves. Consequently, it seems that there probably occurs – at least partly – a transfer of the negative attitude of society towards immigrants and NGOs in general, namely, that people working with immigrants tend to transfer these attitudes onto themselves. We may assume that the media and politicians reinforce such a perception (see Pospěch and Jurečková 2019).

Simultaneously, the research led me to discover that the perceived negative attitude of the public clearly and negatively affects PA work. It regards the overall engagement in advocacy (both the policy and public one), the choice of PA tools, choice of topics and mainly the types of arguments. The interviewees and respondents (in commentaries) admitted that many people working with immigrants close themselves in their social bubbles where they feel safe and comfortable. They focused on the direct work with clients. As for work with the public, IE noted: "*most of [our employees] don't like to speak in public because they know it will bring a shower of insults, name-calling and sometimes threats. But someone usually must perform, and it's usually me. Of course it's unpleasant.*"

It is not only more difficult to organize public campaigns regarding immigrants' rights, but similarly, it is hard to convince politicians to take up migration topics if they were to receive some space in the media. Politicians are afraid that bringing up such topics publicly could endanger or even ruin their political career. As ID assumed, few people would stand for foreigners' rights currently, "*and if they do it, they have to justify it a lot, e.g., by constantly emphasizing that they are talking about those Ukrainians who have been living here for a long time and with whom we have no problems.*"

Consequently, the organizations have to be more careful when choosing the concrete topic to present to politicians and to the public (e.g., immigrant children's needs vs. adults, male needs). Sometimes they attempt to join specific issues relating to a part of the immigrants' issues to wider issues (e.g., women or employees' rights), thus the immigrants' rights would not be in the foreground.

Together with the changing media and politicians' narrative regarding immigrants, the NGOs modify their communication too. As the media and politicians simplify their message about immigration and use shortcut ideas, the NGOs feel forced to do the same. As IA said, *"we have to simplify the messages too, and we can't exactly keep it purely on some of these human rights ethical principles, which [annoy] people . . . It really irritates people when someone at the table says what they should think and what's right."* Thus, the NGOs have to pragmatize and intrumentalize the immigration debate (IA), they have to adapt their argumentation and methods, focus on legal arguments and—to some extent—on security arguments (integration as a security measure). The last one, however, they do not use regularly and are careful with it.

A serious consequence of the atmosphere around immigrants is a tightening up of immigration and integration policies and hindered conditions to address it. Thus, in the last few years, much of the NGOs' work has been regarded as the defense of foreigners' rights, not advancing them. ID admitted: *"Before 2015, 2014 we were talking about the social rights of foreigners, now we are talking about their fundamental rights. Just back."*

The interviewees were aware that the NGOs need to communicate their role better. Their task is to explain not only to politicians but also to the public, why they engage in immigrant integration and why they could be politically relevant, but also what they advocate for, and why. Clarifying their own legitimacy claims and their PA-related accountability is of great importance in this context.

## 4. Discussion and Conclusions

Legitimacy and accountability are questions that the NGOs need to return to frequently, and particularly at present when a great part of society is contesting the acceptance of newcomers or even the presence of immigrants living in the country for a longer time.

The research made it possible to identify a few legitimacy sources for PA, which the NGOs referred to. These are the concepts of civil society and democracy, law—including international commitments, experience and direct contact with clients, as well as moral obligations.

Referring to the specific concept of a civil society and democracy as well as to the law and moral obligations, reveals the NGOs' idea and vision concerning society's functioning. As a source of their legitimacy to raise a voice, it deserves special attention.

One may consider it as proof that the NGOs do not have the legitimacy for their PA, since it is *their own idea* regarding society's functioning. This reminds us of the aforementioned notions of President Klaus, who was concerned about 'NGOism', which he understood as "an ideology that offers an alternative mechanism for public decision-making than the standard, traditional parliamentary democracy based on universal, equal, direct and secret suffrage" (Klaus 2005). Klaus claims that he was not, per se, against NGOs himself being a member or a founder of a few organizations, however, he warned of "the threat of circumventing, replacing, 'bypassing' the institutions of the state" (Klaus 2005). Regardless of the actual Klaus stance on the role of NGOs in a democratic state, it is significant that his statements entered into the public debate and supported the questioning of any engagement of NGOs in political issues. Those, who perceived the role of NGOs only in terms of their service-providing or associating people with a common interest, were asking who to vote for them, so they could attempt to affect the policymaking. Interestingly, some people questioned the political participation of those NGOs that deal with issues of human rights, minorities or the environment. At the same time, the adversaries of NGOs tend to ignore the political engagements of other stakeholders, e.g., economic organizations, including companies, and their lobbyists that are much stronger actors than NGOs. These actors also have *their own visions*.

A more careful look at the NGOs representatives' answers enables us to notice that it is not just the NGOs' *own vision*. This is because, while talking about their idea of society's functioning, they talked about the liberal-democratic regime as well as the legal state (*Rechtsstaat*) in which they live, which has a solid legal basis and which they wanted to

maintain. Democracy and effective law are then *external legitimacy sources, which they have internalized*, and which—together with their experiences with clients, gathered evidence and moral obligations, represent not only a *legitimacy source* but also *the driving force* of their engagement in PA.

The NGOs' understanding of democracy remains close to Dahl's concept, which recognizes the important role of civic engagement for fulfilling the democratic criteria that were mentioned earlier. Dahl observes that one of the modern democracy (polyarchy) characteristics is an effectively enforced right of citizens to form and join autonomous associations that attempt to influence the government. Besides, for a modern democracy it is specific, e.g., access to alternative sources of information, right to a freedom of expression and what is especially important is also criticism of the officials, the conduct of the government, and the implemented policies (Dahl 1989). This is possible thanks to a flourishing civil society, and such a flourishing pluralistic—and politically engaged—civil society is a condition to fulfil the democratic criteria. NGOs represent only a part of civil society but the important one, since their formal organization makes the cooperation and collective action easier. The NGOs as the watchdogs of democracy and the legal state, watch if and how the state respects the law and international commitments. Even though this is 'nothing new', it should be reminded since undermining the role of NGOs weakens democracy.

For the interviewees, their everyday work with immigrants was a special legitimacy source. Thanks to their experience and expertise, they feel it is legitimate that they function as transmission belts—if we accept the term used, e.g., by Albareda (2018)—between immigrants, i.e., inhabitants of the Czech Republic (also partly its citizens), and the policymakers providing the second group with evidence about the functioning and impacts of the policies on the lives of real people. Nonetheless, the immigrants themselves cannot be a legitimacy source because the NGOs do not even *speak with* immigrants in their PA activities, as Slim described it (2002). It is hard for them to do so for the several reasons as pointed above. *Speaking for* or *in favor of* immigrants and simultaneously speaking for the receiving majority is a more appropriate expression (see Slim 2002). It regards, in particular, speaking for the weakest parts of the society, but the NGOs, however, did not consider any of these groups a legitimacy source. We may rather treat them as actors toward which the NGOs feel accountable (see further).

When—after Brown and Jagadananda (2007)—we analyzed the research results from the perspective of the types of legitimacy, we could clearly say that since the NGOs working with migrants meet the regulatory requirements, they enjoy legal legitimacy. Furthermore, the NGOs perceive their normative legitimacy, which they draw from democratic values, norms or standards, not only as written law, but also from the legacies of Czech personalities (Masaryk, Havel, signatories of Charter 77), who are authorities for a large part of the society. As far as pragmatic legitimacy is concerned, the organizations themselves identified the indirect benefits for society that can result from their advocacy for the rights of migrants and society (or at least its weakest parts). Nonetheless, at the same time they admitted that a large part of society was not even aware of this. Finally, cognitive legitimacy is rather the Achilles heel of these NGOs. We may point at several reasons for such a situation. One factor is that a great part of society refuses immigrants, and it also often does not know what the NGOs do in practice., Moreover, top politicians defame them for unfair practices which supports the narrative about them being (refugees/immigrants) 'welcomers' that in the Czech context is not only a distorted but above all a derogatory view. Such a narrative overlaps the narrative that the NGOs develop explaining their positions and attitudes (e.g., in interviews provided for media in the hottest period of the crisis of migration policies). We can also perceive here the shortcomings of the Czech integration policy: although it has been said that informedness, i.e., acquisition of knowledge and information (not only by immigrants, but also by the majority) should be a pillar of this policy, it seems that the public is not aware of the real importance of the NGOs assisting immigrants. Several years ago, the government set itself the task of developing a communication strategy on migration and integration (see Zogata-Kusz 2020). Unfortunately, nothing of the kind has

emerged to date. Such a strategy could be helpful in explaining the role of NGOs as experts and partners of the state in promoting the integration of immigrants.

The research findings correspond to the notion of the NGOs' legitimacy (claims) as derived from 'moral' and 'procedural' accountability, democratic representation and—to some extent—social empowerment (Atack 1999). While talking about legitimacy and legitimation, Schrover et al. (2019) pointed at expert, moral and logistical authority that according to them enables NGOs to play a role in policymaking. The first one means that NGOs know the rules and laws, and have access to statistics. The second one refers to the moral high ground by fighting discrimination or protecting the vulnerable. The third one the authors connect to the NGOs' ability to organise emergency relief fast or mobilise people, with them being regarded, rather, as humanitarian or relief organisations. Nonetheless, in the context of service providers assisting immigrants instead of a logistical authority we could rather talk about professional authority pointing to the fact that the state recognises the ability and professionalism of NGOs in providing services to immigrants and their experience in that sphere. Gnes and Vermeulen (2019) similarly link authority (and credibility) to legitimacy, and they emphasise the importance of the two for understanding the relationship between the organisation and its environment.

The second research question pertains to the accountability for policy advocacy activities, thus the question, who the NGOs assisting immigrants feel accountable to. According to the research findings, the NGOs' representatives focused on *relational accountability*, if we refer to Unerman and O'Dwyer's concept (Unerman and O'Dwyer 2006, pp. 353–56). The list and the order of subjects to whom the NGOs felt accountable to was not as clear as in the case of the legitimacy sources. Remarkably, the interviewees do not feel accountable simply to immigrants. The representatives of the Czech NGOs emphasized that they felt accountable towards immigrants and the society—especially its weakest parts. In this way, they confirmed that their vision of society involves the harmony and integration of all inhabitants. Finally, they referred to their colleagues from other organizations—it was a question of solidarity, co-responsibility and NGO brand building. For this reason, the NGOs discussed intensively their PA activities and their possible impacts on various stakeholders. Interestingly, no one did refer to donors as subjects to whom they felt accountable. In other words, there was not upward accountability—as Naidoo (2004) would call it. We may assume that this was because PA represents only a minor part of these NGOs' activities and they usually do not have a special budget for it (however, even the representatives of the consortium did not refer to their donors). What prevailed then, was a downward accountability to clients and the receiving majority (i.e., both addressees of Czech integration policy). Finally, when referring to one another, the NGOs manifested so-called horizontal accountability (see e.g., Naidoo 2004; Crack 2019). Remarkably, only a few communication partners mentioned specific values towards which they felt accountable (in particular democratic principles). Nonetheless, many of them were referring to values, beliefs and mission as to their legitimacy sources. They were talking about the principles that should rule the society and which they felt obliged to follow. Here we may see some signs of *identity accountability*.

The Czech organizations in particular emphasized how much time they devoted to deciding what topics and how they should arise, and what kind of short-term and long-term impacts the proposed solution could bring to various groups. They discuss things internally, analyzing various options and even arguing as to their positions which are at times far from unanimous. This allows them to come up with balanced proposals to solutions regarding immigrants. They are aware that the policy impacts may similarly affect those who were taking action in favor of change, those who were taking action against it, and those who have remained passive on the specific issue. This recalls the James Q. Wilson's concept of client politics adopted by Freeman in immigration politics. One of Freeman's conclusions was that the types of effects, together with their intensity in various parts of society, differ (Freeman 1995). The Czech NGOs were aware of this and tried to consider various possibilities of effects because they felt a responsibility towards society.

Simultaneously, they noted that the number of factors was great, and that they were just one of the interest groups, i.e., a client—in Wilson's concept (Freeman 1995). Other interest groups are usually more powerful.

These findings oppose the notion of those who consider people from migrant-assisting NGOs as irresponsible welcomers (*sluníčkář*, *vítač* in Czech). The NGOs mainly felt accountable to immigrants *and* the society, or to the society *including* the immigrants.

As for the attitude of the environment towards the NGOs assisting immigrants, most representatives of the NGOs perceived it as negative, even though they often did not come across direct negative reactions towards themselves. The atmosphere clearly affects the NGO advocacy work, namely, their access to authorities, politicians' will to get involved, the raised topics and the tools. Consequently, immigrant-related topics became more suitable for back-door advocacy. In the current situation, NGOs have to take more consideration of the tools and types of arguments that they choose. For instance, they need to be more careful in deciding when and why they should continue to stress value or a principles-led PA and use the human rights approach; or when and why to focus more on evidence-led PA, a more rational approach or even when to use security language and talk about integration as a security measure.

The power-relation is imbalanced and the acceptance of the minimalistic understanding of democracy (only political parties and elections), knocks the argument out of the NGOs' hands. It seems that thirty years of democratization has been not enough to root truly the basic institutions of democracy and democratic processes; it is not enough to understand the importance of civil society for democracy. Limiting the role of NGOs into service-delivering, further questions the role of civil society vis-à-vis the state (mainly pertaining to holding politicians and the state institutions to account). This problem does not regard only Czechia. We should see the situation of the Czech NGOs assisting immigrants in the wider context of the condition of democracy in the region. It resembles to some extent the situation in Hungary, where Orban has developed the anti-NGOs and anti-immigration discourse as part of his fight against George Soros and the Open Society Foundations. In political discourse, humanitarian NGOs became characterized as enemies of the Hungarian nation (Timmer and Docka-Filipek 2018). In neighboring Poland, where the ruling parties won the elections also with anti-immigration slogans, these NGOs have not been criminalized, nonetheless, they have remained cut off from European funding and have had to struggle to survive (Klaus et al. 2017). Consequently, the way of addressing the migration crisis on the Polish-Belorussian border similarly raises concerns regarding the quality of Polish democracy.

The research results we may see in the context of migration governance, despite communicative partners did not mention this. The governance, i.e., the process of governing involving a plurality of state and non-state actors, public and non-public, has become one of the basic paradigms for policymaking in European countries. It creates opportunities for increasing the role of NGOs in policymaking. There is a growing literature on migration governance from which it is evident that there is a space for NGOs to be recognized and involved as actors in policymaking at various levels (e.g., Zincone and Caponio 2006; Geddes 2018; Caponio et al. 2018; Weinar et al. 2019; Panizzon and Riemsdijk 2019; Balaz and Cemova 2019). Despite that there is evidence that the role of NGOs in policymaking—in particular at the national level—has been limited (Spencer 2017), it is still valid what Brown and Jagadananda (2007) argue, that—given the growing roles of NGOs in various levels governance—improving civil society legitimacy and accountability is of critical significance.

## 5. Final Remarks

Policy advocacy has been a relatively new activity in Czech civil society organizations. Most NGOs have only just begun learning how to utilize their potential in affecting policies. Politicians are not always ready to cooperate with a third sector—more so when a large part of society often has a distorted understanding of what the NGOs do and why. This does not regard only anti-immigrant concerns in the society, but also the misunderstanding

of the role of NGOs in policymaking, their motivation and capacities. Nonetheless, we must assume that NGOs—in general, not just those assisting immigrants—are partly co-responsible for their current situation. They have not been communicating their role enough during recent years. A positive outcome is that recently they have been cooperating on a campaign whose aim is to support trust in NGOs and to fight disinformation about them with data and facts. The 2021 campaign, *"That's who we are"*, is to show their diversity, contribution to society and to the state, to explain the ways they gain and spend finances and to highlight the values of a free society.

If there is a lack of understanding of the NGOs' functioning in society, it is relatively simple to perceive that eventually some will make attempts to corner the NGOs into the role of voiceless service-delivers. Although the NGOs refer to their own vision of society when talking about their PA-related legitimacy, the research revealed that this is a manifestation of internalized external legitimacy sources such as democratic principles already enshrined in the constitution and in existing laws. Therefore, only advocating for immigrant rights, while omitting the rights of the weakest parts of the receiving society, is not a perception that fits with the NGOs' vision of society. Therefore, they feel accountable to both immigrants and to (mainly the weakest parts of) society.

**Funding:** This research was funded by IGA_CMTF_2019_010 and IGA_CMTF_2021_007.

**Institutional Review Board Statement:** Ethical review and approval were waived for this study, due to informed consent obtained from all subjects involved in the study.

**Informed Consent Statement:** Informed consent was obtained from all subjects involved in the study.

**Data Availability Statement:** Research data are available from the authors upon request.

**Acknowledgments:** I would like to thank to Hana Šlechtová for critical feedback on the text of the article.

**Conflicts of Interest:** The author declares no conflict of interest.

## Notes

[1]     Charter 77 was the document that criticized the Czechoslovak government for failing to implement the human rights provisions included in documents it had signed. Many of its signatories were persecuted. After 1989, a few of the signatories of Charter 77 founded NGOs assisting immigrants.

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
