# Peer review of "Policy Advocacy and NGOs Assisting Immigrants: Legitimacy, Accountability and the Perceived Attitude of the Majority"

_socsci, doi:10.3390/socsci11020077_

Round 1
Reviewer 1 Report
This is a well written study that is clearly structured and that draws from an impressing material. The findings of it adds to the current literature. Although my impression is generally positive, I have a few minor remarks that I would recommend to be addressed.
- The author(s) draws on an impressing material. This is obviously a merit of the paper but it is partly unclear how the different data sources are integrated with each other. Hence, I would encourage the author to look closer into the mixed methods literature to guidance on how to think about the combination of data and methods.
- In the discussion and conclusions I think that the author could try to lift the argumentation to more of an theoretical level. One way could be to relate findings to the substantial literature in migration studies. Another way could be to expand discussions beyond the empirical case and also reflect how findings potentially could be generalized.
- Lastly a few additional comments. I believe that the author could strengthen the passages concerning the negative narrative that have been associated with immigration in many countries. See for example Boréus “Migrants & Natives – them and us”. Furthermore, I think that the author could use a bit more informative headlines than now included (“Research”, “Results”).
Author Response
Dear Reviewer,
I am very grateful for the time you spent with my article and for your valuable comments. I reviewed my article according to your suggestions.
For integration of the research methods (data sources): see rows 328-338;
For lifting the argumentation to more of an theoretical level and relating findings to the literature in migration studies: see rows 753-767 and 841-853;
For the negative narrative associated with immigration in many countries: see rows 261-270 and 833-835;
The headline Research I have changed to: Four stages multi-method research.
As for the other headlines, I would leave them as they are – I am afraid that making them more informative may narrow the information about their content too much. Besides, I as I see from other articles in Social Sciences, my article will not be exceptional in this regard (see e.g. https://www.mdpi.com/2076-0760/11/2/55/htm ).
Please see the attachment.
I am looking forward you further comments.
Thank you very much.
With kind regards,
the author of the article
Reviewer 2 Report
The topic is not new but offers a Czech perspective, which can be useful for expanding and enriching the debate.
I believe some aspects are missing in the article:
- in the part about the Czech context, it can be added something about the NGOs which are focusing in migrants (their tasks; their main activities, etc);
- there is a quite ample debate on relations with political parties. Some more details on the political situation in the country could be useful to the reader;
- finally, more details on the questionnaire could be useful to the understanding of results.
Author Response
Dear Reviewer,
I am very grateful for the time you have spent and you will spend with my article and for your valuable comments. I reviewed my text according to your suggestions:
For the NGOs tasks and activities: see rows 229-246;
For the political situation: see rows 199-209;
For more details on the questionnaire: see rows 329-353 (some of these lines is the reaction for the comments of the other reviewer).
Please see the attachment.
I am looking forward you further comments.
Thank you very much.
With kind regards,
the author of the article